# DOES RESISTANCE TO STYLE-TRANSFER EQUAL GLOBAL SHAPE BIAS? MEASURING NETWORK SENSITIVITY TO GLOBAL SHAPE CONFIGURATION

**Ziqi Wen,**[*] **Tianqin Li**[*] **Zhi Jing & Tai Sing Lee** [†]
School of Computer Science
Carnegie Mellon University
Pittsburgh, PA 15213, USA
ziqiwen, tianqinl, zjing2, taislee@andrew.cmu.edu

## ABSTRACT

Deep learning models are known to exhibit a strong texture bias, while human tends to rely heavily on global shape structure for object recognition. The current benchmark for evaluating a model's global shape bias is a set of style-transferred images with the assumption that resistance to the attack of style transfer is related to the development of global structure sensitivity in the model. In this work, we show that networks trained with style-transfer images indeed learn to ignore style, but its shape bias arises primarily from local detail. We provide a **Disrupted Structure Testbench (DiST)** as a direct measurement of global structure sensitivity. Our test includes 2400 original images from ImageNet-1K, each of which is accompanied by two images with the global shapes of the original image disrupted while preserving its texture via the texture synthesis program. We found that (1) models that performed well on the previous cue-conflict dataset do not fare well in the proposed DiST; (2) the supervised trained Vision Transformer (ViT) loses its global spatial information from positional embedding, leading to no significant advantages over Convolutional Neural Networks (CNNs) on DiST. While self-supervised learning methods, especially mask autoencoder significantly improve the global structure sensitivity of ViT. (3) Improving the global structure sensitivity is orthogonal to resistance to style-transfer, indicating that the relationship between global shape structure and local texture detail is not an either/or relationship. Training with DiST images and style-transferred images are complementary and can be combined to train networks together to enhance the global shape sensitivity and robustness of local features. Our code will be hosted in github: `https://github.com/starsky77/DiST`

## 1 INTRODUCTION

Deep learning models for object recognition are known to exhibit strong texture bias (Geirhos et al., 2018; Baker & Elder, 2022). In solving problems, neural networks tend to discover easy shortcuts that might not generalize well (Ilyas et al., 2019; Drenkow et al., 2021). Rather than learning a more structured representation of objects, i.e., the global configuration of the local components, a.k.a. global shape structure, or **global structure**, convolutional neural networks trained for classifying objects rely primarily on the statistical regularities of features discovered along the network hierarchy. Standard networks fumbled badly when the test images were subjected to style or texture transfer (Geirhos et al., 2018), revealing their reliance on texture and local feature statistics, perhaps the easiest features, rather than global structure. Humans, on the other hand, are fairly robust against such style transfer manipulation in object recognition, indicative of our explicit utilization of global shape structure (Ayzenberg & Behrmann, 2022; Ayzenberg & Lourenco, 2022; Quinn et al., 2001a;b). Such preferences for the global shape structure of the object are so-called global shape bias.

---

[*]Denotes equal contributions
[†]Corresponding Author

**(a) Feature Attribution Results**   **(b) Our Disrupted Structure Testbench**

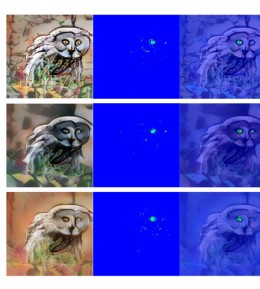 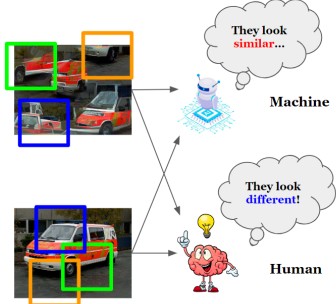

Stylize Augmented Models          DiST directly shows models'
still focus on **local feature**       sensitivity of **global shape**

Figure 1: **Left**: Feature Attribution Analysis based SmoothGrad (Smilkov et al., 2017) on stylized augmentation trained models. Surprisingly, models that can resist style transfers still be primarily sensitive to local features, rather than the global shape configuration. **Right**: Illustration of our proposed Disrupted Structure Testbench (DiST). We hope machine would successfully distinct the images that have disrupted global structure from the original image, align with human that are using the global shape structure as a cue for object recognition

To measure how well models understand the global shape structure, the style-transferred images, specifically, the image that transfers its texture into an image that belongs to a different class, known as the cue-conflict images, are commonly used as the benchmark of global shape bias (Vishniakov et al., 2024; Geirhos et al., 2021). Various approaches have been developed to steer neural network learning towards shapes from texture (Brochu, 2019; Geirhos et al., 2021; Li et al., 2023). Based on such a way of measurement, the most effective approach remains to be augmenting the training data with randomized style-transfer operation (Geirhos et al., 2018; 2021).

Despite the common usage of cue-conflict images, a crucial hypothesis underlaying the cue-conflict benchmark is that: *if the models are not relying on the local texture, then it relies on the global structure*. However, in this paper, we found that even if models are forced to be robust against style-transfer operation through such augmentations, models are still finding a shortcut that is not global shape structure. In Figure 1, we can see that a stylized trained neural network becomes resistant to the style changes, however, its sensitivity map still shows heavy focus on the **local features** (the eye of the owl in this case), rather than the **global structure** (See results of *Feature Attribution Results* in Figure 1(a), bright pixels in the middle column indicates the area is sensitive to the model perception, we refer the details to Section A.1). Our experiments show the relationship between global shape structure and local texture is not an either/or relationship, suggesting that showing the model is resistant to the change of texture does not necessarily means it understand the global shape structure.

To remedy this problem, we developed evaluation dataset, called **DiST (Disrupted Structure Testbench)**, to directly evaluate the sensitivity to the global shapes structure (Figure 1(b)). In this dataset, images were transformed to disrupt their global structure while maintaining their texture statistics. We used DiST to perform an odd-man-out test on the various models and human subjects to measure their ability to distinguish the original image from its structure-disrupted variants as a metric of their global structure sensitivity. We found humans far superior to Style-transfer-trained networks in discriminating the differences in global forms. In fact, the Style-transfer network's performance is no better than the standard CNN that has not been subjected to augmented Style-transfer training.

Besides, based on DiST, we are able to discover several surprising findings that has not been shown by cue-conflict benchmark. Vision transformers (ViTs) with supervised learning fared no better than standard CNN, contradicting the beliefs that ViTs had captured and utilized the global relationship of object parts in this task. While the self-supervised learning (SSL) method that uses masked autoencoder (MAE) significantly improve the global structure sensitivity of ViT.

Along with the DiST dataset, we also demonstrate that networks trained with augmented DiST data also do well in discriminating the global structure of objects if using a carefully designed training approach (we name the method as **DiSTinguish**). Finally, we found that the DiSTinguish-trained

network and Style-transfer-trained network are orthogonal and complementary, as one focuses on global structure, while the other tends to capture robust local feature. Thus, our paper provides a better and deeper understanding of the nature of global shape bias and texture bias within the networks. Moreover, through such evaluation method we are able to find self-attention and positional embedding itself does not necessarily provide perception of global structure, while how the model is trained is the thing that matters.

## 2 RELATED WORK

Deep Neural Networks (DNNs) have been the cornerstone of the revolution in computer vision, delivering state-of-the-art performance on a wide array of tasks (Luo et al., 2021; Redmon et al., 2016; He et al., 2016; Brown et al., 2020). However, understanding DNNs has been a vital topic to further advancement of these black box models (Drenkow et al., 2021; Petch et al., 2022; Gilpin et al., 2018). One aspect of understanding DNNs in vision systems is identifying the biases they might have when classifying images. Two prominent visual cues are local texture detail and global shape structure (Garces et al., 2012; Janner et al., 2017).

Originally, it was believed that DNNs, especially those trained on large datasets like ImageNet, primarily learn shapes rather than textures, as visualization in convolutional neural networks shows clear hierarchical composition features of various levels of object shapes (Zeiler & Fergus, 2014). This belief was also based on the intuitive understanding that shape structures are more semantically meaningful than textures for most object categories. However, Geirhos et al. (2018) challenged this belief and showed that DNNs trained on ImageNet have a strong bias towards texture. Our work re-examines their proposed style-transfer-based approach and further checks if the model has truly understood the global shape structure of the image.

Geirhos et al. (2021) benchmarked various widely used models on the proposed Style-transfer datasets in Geirhos et al. (2018). Among these, network architectures play a significant role in improving the models' global shape bias. Compared to the convolutional neural networks (CNNs), the newly proposed vision transformer family ViTs (Dosovitskiy et al., 2020) perform significantly better in terms of the Style-transfer based shape bias as well as other corruption-based robustness test measured by Paul & Chen (2022). However, we observe that supervised trained ViTs yield no significant improvement on our proposed global structure disrupted test, the spatial information of patches is lost as the feedforward continues. While the ViT trained with self-supervised learning (SSL), (e.g. DINO (Caron et al., 2021)) are significantly better, especially masked autoencoder (He et al., 2022), which shows even better performance than human.

## 3 METHODS

### 3.1 DISRUPTED STRUCTURE TESTBENCH (DIST)

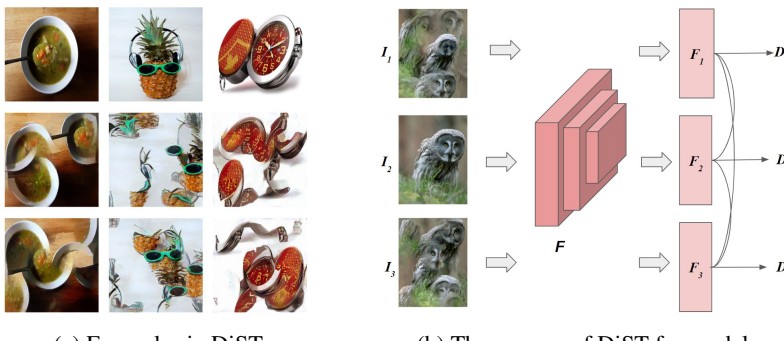

(a) Examples in DiST      (b) The process of DiST for models

Figure 2: Disrupted Structure Testbench (DiST)

Our Disrupted Structure Testbench (DiST) deliberately compares the representation before and after we disrupt the global structure of the image. One could imagine there could be many random global

structure disrupted variants of an original image as the joint spatial configuration of local parts could be arbitrary. To get a quantitative measurement of the models' global structure sensitivity, DiST formulates the evaluation metrics as the accuracy of an oddity detection task. The subjects to DiST (models or humans) are asked to select a distinct image from a pool of choices, which consists of one original image where the global structure is intact, and $N$ global structure disruption variants of the original image (each of which preserves the local patterns) . We pick $N = 2$ for all the DiST tests as we observe that increases $N$ will not increase the difficulty of the task.

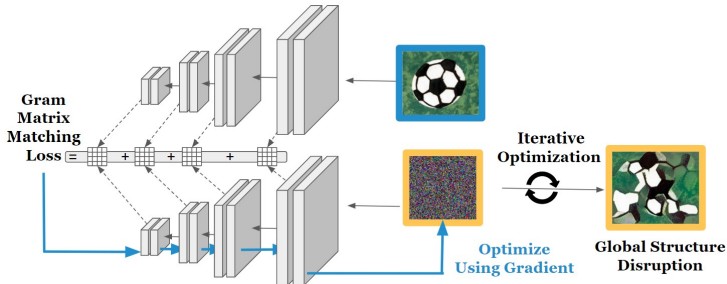

Figure 3: Mechanism of computing global structure disruption images. We implement approach proposed in Gatys et al. (2015). Specifically, we optimize a randomly initialized image (yellow) so that when it passes through a pretrained VGG network, its intermediate layers' gram matrix match the targeted image (blue). This results in preserving the images' local features but randomizing the global structures.

**Structure Disruption via Texture Synthesis Program**    Texture Synthesis allows for the generation of images that retain the original texture details while randomizing the global structures. We utilize Gatys et al. (2015) in particular to construct the global structure disrupted images for the DiST oddity detection task. We illustrate the process of this texture synthesis in Figure 3. For any given target image, $I_t \in R^{(3,H_0,W_0)}$ (blue boundary image in Figure 3), we want to get $I_o \in R^{(3,H_0,W_0)}$ that possess the same local features but disrupted global structure (as shown in yellow boundary images on the right). To achieve this, we initialize tensor $I_1 \in R^{(3,H_0,W_0)}$ using value independently sampled from an isotropic Gaussian distribution and complete the process of $I_1 \rightarrow I_2 \rightarrow I_3... \rightarrow I_o$ through minimizing the $L$ as Gram Matrix Matching Loss. Specifically,

$$\frac{\partial Loss}{\partial I_i} = \frac{\partial}{\partial I_i} \sum_l ||\text{Gram}(A_l(I_t)) - \text{Gram}(A_l(I_i))||^2 \tag{1}$$

where $A_l(I) \in R^{(C_l,H_l,W_l)}$ denotes the $l$-th layers activation tensor from which we destroy the global spatial information by computing the channel-wise dot products, i.e. $\text{Gram}(A_l(I_t)) \in R^{(C_l,C_l)}$.

**DiST Metric Formulation**    For each trial in the Disrupted Structure Testbench (DiST), two global structure disrupted variants are generated using distinct random seeds. Each image, denoted as $I_i$, is then passed through the evaluation network $F$ to obtain a feature vector $F(I_i)$ from the final layer. The model identifies the image most dissimilar to the others by calculating the cosine distance between feature vectors. The procedure for this calculation is as follows:

$$D_i = \sum_{j \neq i} (1 - \frac{F(I_i) \cdot F(I_j)}{||F(I_i)||_2 ||F(I_j)||_2})/N \tag{2}$$

$N$ represents the number of structure-disrupted images in each trial, in DiST it would be equal to 2. The dissimilarity of the image $I_i$ to the other 2 images is calculated as the average of the pairwise cosine distance of each two image pair. Cosine distance of vector $u$ and $v$ is calculated as $D_C(u,v) = 1 - S_C(u,v)$, where $S_C(u,v)$ is the cosine similarity.

The model will select the images $a$ that is the most different from the other two images based on $D_i$: $a = \arg\max_i \frac{exp(D_i)}{\sum_j exp(D_j))}$. DiST is fundamentally different from the evaluation methods based

on style transfer or other changes in texture details. Those methods apply style transfer operation to generate the evaluation data. The model trained with stylized augmentation could get an advantage in that evaluation due to the familiarity with different style domains. In contrast, DiST involves no style transfer operations. Instead, it directly assesses the representations learned by the model to show how sensitive it is due to the change in global structure. This approach eliminates any biases arising from familiarity with stylized images, offering an entirely new angle from which to evaluate global shape bias.

### 3.2 PSYCHOPHYSICAL EXPERIMENTS

Human vision is known to exhibit a strong bias towards shape. To quantify the gap between deep learning models and the human visual system, we conducted a psychophysical experiment with human subjects. To align this experiment closely with deep learning evaluations, participants were simply instructed to select the image they found to be "the most different," without receiving any additional hints or context. To mimic the feedforward processes in deep learning models, we displayed stimulus images for a limited time, thereby restricting additional reasoning. Furthermore, participants received no feedback on the correctness of their selections, eliminating the influence of supervised signals.

In each trial, participants were simultaneously presented with three stimulus images for a duration of 800 ms. They then had an additional 1,200 ms, making a total of 2,000 ms, to make their selections. Any response given after the 2,000-millisecond window was considered invalid. To mitigate the effects of fatigue, participants were allowed breaks after completing 100 trials, which consisted of 100 sets of images. We accumulated data from 16,800 trials and 32 human subjects to calculate the overall performance of DiST to represent the human visual system. The final results, shown in Fig.5, represent the average performance across all participating human subjects. Further details of the psychophysical experiments and how the experiment is conducted can be found in the appendix.

### 3.3 DISTINGUISH BETWEEN THE ORIGINAL STRUCTURE AND DISRUPTED STRUCTURE

Deep learning models are excellent learners when we explicitly define the learning objectives. Stylized augmentation forces the model to learn style-agnostic representation, leading to its impressive performance in the stylized domain. Here we would like to directly force the model to distinguish between the original shape and the disrupted one. We propose **DiSTinguish**, as shown in Fig.4, a simple supervised training approach to explicitly enforce the constraint to guide the model to learn the global structure of the object. Rather than operating within the confines of an $n$-class classification task, we expand this to $2n$ classes. Where the loss of the network would be: $L(\theta) = -\sum_{i=1}^{2n} y_i \log(p_i)$, where $p_i$ is the predicted probability of the sample belonging to class $i$ and $y$ is the one-hot code for the ground truth label of $2n$ classes. This expansion incorporates structure-disrupted versions of each original class as additional, separate classes. As it's not practical to produce the full structure-disrupted version of ImageNet1K, we applied an approximation here, the details of the approximation methods and their effectiveness are shown in the appendix.

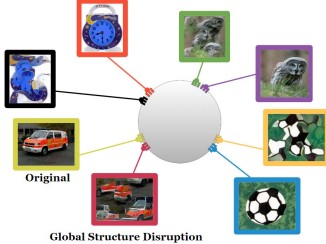

Figure 4: DiSTinguish Training, structure-disrupted images are added as separated classes

During the evaluation phase, the model reverts to $n$-class classification by summing the logits corresponding to the disrupted structure and original classes: $z_i' = z_i + z_{i+n}$, where class $i$ and class $i + n$ are the original structure and its structure-disrupted counterpart, to reduce the output logits $Z$ from $2n$ dimension to $n$ dimension. While for DiST, this remapping is unnecessary, as we directly compare the feature vectors learned by the model rather than the classification logits.

## 4 RESULTS

### 4.1 GLOBAL STRUCTURE SENSITIVITY: MODEL V.S. HUMAN

Cue-conflict dataset (Geirhos et al. (2021)) directly tests the model's robustness against style-transferred operation. It tests the model's classification accuracy on the style-transferred images. The score of the Cue-Conflict dataset is determined by calculating the proportion of instances where the model correctly classified as the image's shape, rather than texture, which is defined as $\frac{\text{Number of Correct Shape Recognitions}}{\text{Number of Correct Recognitions}}$.

Although the result can clearly show if the model is relying on texture detail, to claim that the model is making use of global shape structures, it is based on the hypothesis that the model either uses the texture detail or uses the global structures, and there is no third option. If such a hypothesis holds: which means resistance to texture change is equal to using more global structure information, then the trend on DiST accuracy and cue-conflict score should be similar, as DiST is designed to directly measure the global structure sensitivity.

To determine whether this hypothesis holds, we use DiST to directly evaluate the global structure sensitivity of the models across different architectures and different training methods. Those models include: **transformer architectures** (e.g. ViT, DeiT (Touvron et al. (2022)) and ConvNeXt (Woo et al. (2023))), **traditional CNN architectures** (e.g. ResNet, ResNeXt (Xie et al. (2017)), Inception (Szegedy et al. (2017)), DenseNet (Huang et al. (2017))). **Mobile network** searched by neutral architecture search (e.g. MNasNet (Tan et al. (2019), MobileNet (Koonce & Koonce (2021))). We also cover models trained with different techniques, those technique used to show significant effectiveness on improving model's global shape bias based on the cue-conflict dataset, including **adversarial training**, **sparse activation** (Li et al. (2023)) and **semi-supervised training** (Xie et al. (2020)). As shown in Fig.5, where both human and model performances are ranked according to DiST Accuracy, we highlight the important discoveries as follows:

Firstly, ResNet50-SIN, which is trained on stylized images (Geirhos et al. (2018)), outperforms other models on the Cue-Conflict dataset but fails to surpass the performance of a normally trained ResNet50 on DiST (52.6% v.s. 69.4%). This suggests that its high performance on the Cue-Conflict dataset may not be attributed to a better understanding of the global structure but rather to some other learned "short-cut". Secondly, the DiST Accuracy of the models is inconsistent with their Cue-Conflict score. This suggests that not relying on texture detail (leads to high Cue-Conflict score) does not necessarily mean using the global structure information (leads to high DiST Accuracy).

For the human evaluation result, we average human performance data collected from 16,800 trials to obtain the final human performance metric, which is 85.5%, outperforming almost all the deep learning models. The results consistently show that humans are robust shape-based learners, irrespective of the evaluation method used.

### 4.2 THE LOSS OF GLOBAL SPATIAL INFORMATION DURING SUPERVISED LEARNING OF VIT.

The core of vision transformer (ViT) (Dosovitskiy et al., 2020): self-attention layer along with the positional embedding are believed to be more expressive than the convolution operation in terms of capturing global structure. However, as depicted in Fig.5, such an idea is challenged by the result that ViT models haven't exhibited a significant advantage over ResNet when evaluated on DiST. Notably, ViT-B even underperforms compared to ResNet50. Additionally, although the performance on the Cue-Conflict dataset suggests that a large model size is more robust against texture changes, a larger model size does not guarantee enhanced capability in perceiving global structures. The performance of various ViT sizes on DiST is incongruent with their parameter sizes.

We further evaluate the DiST accuracy of the same ViT architecture with different training methods: supervised learning, compared with self-supervised learning (SSL): DINO (Caron et al., 2021), and Masked Autoencoders (MAE) (He et al., 2022). As shown in Table.1. ViT trained with SSL has shown significantly better performance on DiST compared with supervised learning. Surprisingly, with 93.7% accuracy, MAE is even better than human performance.

To understand why supervised training failed to capture global structure as well as MAE, even if positional embedding should have provided information on the global spatial relationship of each

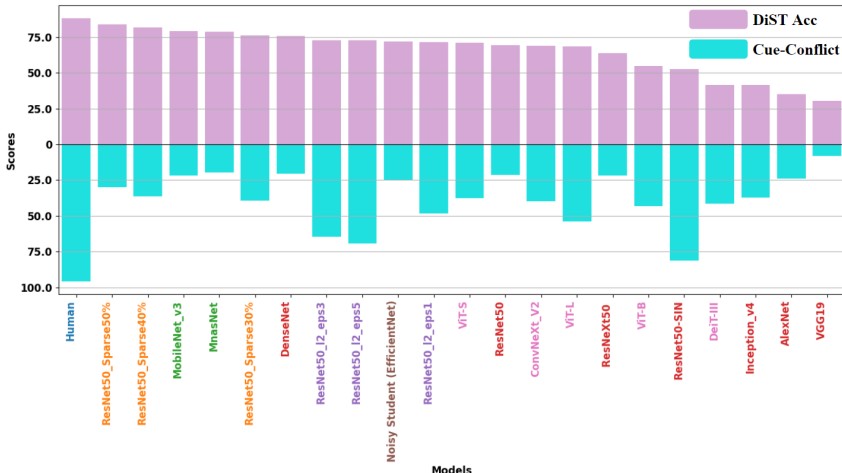

Figure 5: Human and different models' performance on DiST and Cue-Conflict dataset.

local component, we investigate if the spatial information is still encoded in the final representation of ViT with different training methods.

Specifically, we train a 2-layer Multi-Layer Perceptron $f_\theta(v)$ that takes in only the embedding vector $v$ of a certain image patch at a specific layer and outputs the prediction of its spatial location $(x, y)$. We measure how well the network decodes via normalized regression errors $(0.5 * (|f_\theta(v)_x - x|/\texttt{max}(x) + |f_\theta(v)_y - y|/\texttt{max}(y))$. This decoding regression test reflects how much information is encoded inside the patch representation in certain layers. If the global spatial information is still kept within the embedding, the network should easily decode the location correctly with a simple training process. Here we describe the data for this regression task in detail: Imagine we have $N \times L \times (D + 2)$ for a certain layer's intermediate representation, with $N$ denoting total number of images, $L$ denotes the sequence length of the transformer, and $D$ representing the dimension of each individual token embedding at that layer and 2 is the $(x, y)$ value for each image patch. Then we construct our input data as $(N * L) \times D$ and our corresponding regression target as $(N * L) \times 2$. A 5-fold cross-validation is performed. The results are shown in Fig.6a. We repeat the above mentioned test for each layer inside the transformer.

The results indicate that as the layer goes deeper, it becomes harder for the simple network to decode the correct position of the image patches. Notably, it becomes significantly harder for supervised trained ViT to decode the correct position based on the patch embedding compared with ViT trained with SSL. Consist with the result shown by DiST, using the embedding trained with MAE, the simple network can easily decode the position of the patch, indicating that spatial information stored in positional embedding is still maintained as the layer goes deeper, while such information is easy to lose with supervised learning. The attention map of the final layer of ViT in Fig.6b further shows that ViT trained with MAE tends to have more global attention that covers the whole object, even if the structure has been disrupted, such attention still works globally, while the head of supervised trained ViT focused on local part on both two versions of images.

Table 1: ResNet50 and different ViTs' performance on DiST and Cue-Conflict, as SSL methods do not have a classification head, the pretrained model cannot be directly tested on Cue-Conflict

| Model | # Param (M) | Cue-Conflict score(%)(↑) | DiST Acc(%) (↑) |
|---|---|---|---|
| ViT-L | 303.3 | 53.8 | 68.5 |
| ViT-B | 85.8 | 43.1 | 54.8 |
| ViT-B (DINO) | 85.8 | - | 74.9 |
| ViT-B (MAE) | 85.8 | - | 93.7 |
| ViT-S | 22.1 | 37.7 | 70.9 |
| ResNet50 | 25.5 | 21.4 | 69.4 |

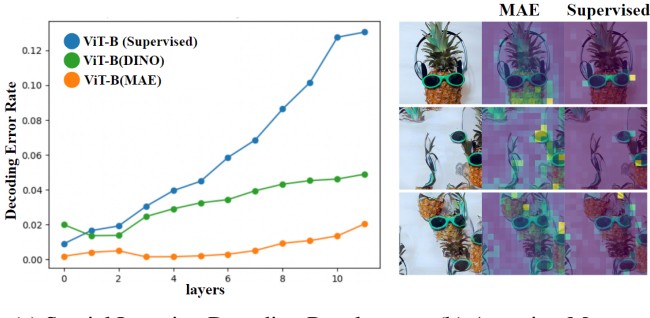

(a) Spatial Location Decoding Result       (b) Attention Map

Figure 6: **Left**: Decoding results of using the embedding of each image patch to predict the the 2D coordinates of the patch, as the layer goes deeper, it becomes hard for supervised ViT to correctly decode the location. **Right**: The Attention Map for [*CLS*] token in the last layer of ViT trained with different methods on Normal and Structure Disrupted Image

## 4.3 THE ORTHOGONALITY OF USING GLOBAL STRUCTURES AND RESISTANCE TO STYLE-TRANSFER

As augmented with style-transferred images force the model to be robust against texture changes, we follow a similar idea to force the model using global structure information. We employ DiSTinguish to explicitly train the model to differentiate between original and disrupted structures, and compare its effect with style-augmentation technique.

We evaluate a ResNet50 model under four distinct training approaches. **(i) Baseline:** The model is trained using pre-trained weights without any specialized augmentation. **(ii) Stylized Augmentation:** We employ AdaIN (Huang & Belongie (2017)) to create stylized versions of the ImageNet1K dataset. Each class receives an additional 100 augmented images. **(iii) DiSTinguish:** An extra 1000 structure-disrupted image classes are created, each containing 100 images, and the model is trained as part of a 2000-class classification task. **(iv) DiSTinguish + Stylized Augmentation:** Combining DiSTinguish and Stylized Augmentation together, where the model will trained with 2000-class classification task, and the original 1000 classes images would also contain the stylized images as augmentation. Except for the pre-trained Baseline model, where we directly use the IM-AGENET1K_V1 weights, all other models are trained under identical configurations. We evaluate the above four methods on three different evaluation datasets: DiST, a style-transferred version of the evaluation dataset of ImageNet1K (SIN-1K), and the original evaluation dataset of ImageNet1K.

As shown in the Table.2, DiSTinguish significantly enhances performance on the DiST evaluation while maintaining comparable results on the original dataset. Importantly, the effectiveness of DiS-Tinguish is orthogonal to the effectiveness of style augmentation, as those two specially designed augmentation techniques do not significantly influence the performance on the benchmark created with a different technique. Moreover, DiSTinguish is fully compatible with stylized augmentation techniques, which allows for their combined use without any performance degradation in either the stylized domain or the DiST evaluations.

We further examine the feature vectors from the last layer of a ResNet50 model trained under different conditions by visualizing them using t-SNE. As illustrated in Fig.10, both the baseline model and the one trained with Stylized Augmentation fail to effectively distinguish between original and disrupted structure. Their corresponding classes in the feature space show significant overlap. In contrast, the model trained with DiSTinguish clearly separates feature clusters corresponding to the original structure from those of their disrupted versions. And combining two methods together won't influence such separation.

## 4.4 WHY STYLIZED AUGMENTATION FAILED

To investigate why Stylized Augmentation fails to significantly improve DiST and how DiSTinguish achieves better performance, we employ SmoothGrad (Smilkov et al. (2017)) to generate sensitivity maps for a ResNet50 model trained using either DiSTinguish or Stylized Augmentation. Sensitivity maps reveal how responsive the model is to changes in pixel values. To clearly show which regions

Table 2: DiSTinguish and Stylized Augmentation's performance on ImageNet1K

| | ImageNet1K (↑) | | SIN-1K (↑) | | DiST (↑) |
|---|---|---|---|---|---|
| | Top-1 | Top-5 | Top-1 | Top-5 | |
| Baseline | 76.1 | 94.0 | 26.1 | 47.6 | 69.4 |
| Stylized Aug | 78.1 | 94.1 | 52.2 | 75.2 | 73.3 |
| DiSTinguish | 77.7 | 93.8 | 24.9 | 44.7 | 98.6 |
| DiSTinguish + Stylized Aug | 77.8 | 94.0 | **52.2** | **75.7** | **98.7** |

contribute most to the model's internal representation, we use a binary mask to mask out the pixels that have low sensitivity values.

Across various stylized images, models trained with Stylized Augmentation tend to focus on specific local features that remain relatively invariant to changes in style. For instance, as depicted in the row (ii.), (iii.), and (iv.) in Fig.7, a stylized augmented model may rely heavily on a single eye as the key feature for its decision-making. We hypothesize that the feature associated with the eye remains stable even when the style domain undergoes significant alterations. This enables the stylized augmented model to classify the image correctly despite changes to many texture details. However, this strategy fail in the DiST evaluation, where the global structure is altered but local features remain constant. As illustrated in row (i.) of Fig.7, the stylized augmented model fails to account for the global structure, concentrating solely on distinctive features like eyes and neglecting other regions.

In contrast, models trained with DiSTinguish are compelled to make use of global features to effectively differentiate structure-disrupted images from original ones. Consequently, the model's sensitive regions are not confined to small, local areas; rather, they extend to larger, global structures. As shown in row (i.) of Fig.7, the model is highly responsive to most parts of the owl, even when they are spatially separated, thus enabling it to perceive changes in the global structure. This sensitivity to global features persists even in style-transferred images. Compared to stylized augmented models, which fixate on specific local features such as an eye, models trained with DiSTinguish are sensitive to the entire object. This suggests that the two training methodologies engender fundamentally different feature preferences in models. While DiSTinguish encourages models to focus on global structures to discern between disrupted and original structure, Stylized Augmentation prompts the model to rely on features that remain stable across various style domains as a defense against style transfer operations.



Figure 7: Sensitivity map of ResNet50 trained with DiSTinguish or Stylized Augmentation

## 5   CONCLUSION

We introduced the Disrupted Structure Testbench (DiST) as a direct metric to evaluate whether the model has understood global shape structure. Based on our proposed methods, we have revealed three key insights: (i) Existing models acclaimed for shape bias perform poorly on DiST. (ii) Supervised trained ViT does fully capture the spatial information from positional embedding, while masked autoencoder successfully keeps it. (iii) Not relying on texture detail is not equal to using global structure information. Forcing the model to ignore texture detail is complementary and orthogonal to forcing the model to learn global shape structure.

## 6  ACKNOWLEDGEMENT

This work was supported by an NSF grant CISE RI 1816568 and NIH R01 EY030226-01A1 awarded to Tai Sing Lee. This work is also partially supported by the graduate student fellowship from CMU Computer Science Department. We thank Junru Zhao for helpful discussion. We also thank NSF ACCESS Allocations (Project Number CIS230221) for computational resource support.

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

# A  Appendix

## A.1  Feature Attribution Analysis

The feature attribution analysis is done by using smoothGrad (Smilkov et al. (2017)), one of the gradient-based sensitivity maps (a.k.a sensitivity maps) methods that are commonly used to identify pixels that would strongly influence the decision of the model. Specifically, gradient-based sensitivity maps try to visualize the gradient of the class-predicted probability function with respect to the input image, which is $M_c(I) = \partial F_c(I)/\partial I$, where $F_c$ is the function that predicts the probability that input image $I$ belong to class $c$.

Traditional methods for computing sensitivity maps often suffer from noise, making them difficult to interpret. SmoothGrad improves the quality of these maps by taking the average of the gradients obtained by adding noise to the input multiple times and recalculating the gradient for each noisy version. Specifically, it generates $N$ noisy versions of the input $I$ (e.g. Gaussian noise). For each noisy input, perform a forward and backward pass through the neural network to compute the gradient of the output with respect to each input feature. Then average the gradients across all noisy inputs to create a clear sensitivity map.

During our analysis, to visualize the sensitivity map, sensitivity scores are rescaled to fall within the range of $0$ and $1$. To further clarify the regions of sensitivity, we apply a threshold to create a binary mask based on these scores. In the main experimental context, this threshold is set at $0.15$. Original sensitivity maps without the binary mask will also be presented in the following section for comparison.

## A.2  10-step optimization as an approximation during DiSTinguish

Due to the high time cost of the texture synthesis process(100 optimization steps would take above 55s on a single Tesla V100 GPU, which is the time cost for a single image generation). It's infeasible to generate a full structure-disrupted version of the ImageNet1k training dataset. Therefore, we use 10-step optimization results to approximate the 100-step optimization result in DiST, each additional structure-disrupted class would have 100 images. The effectiveness of this approximation is shown through the following small-scale experiments.

We select 10 classes from ImageNet1K. The choice of the classes is the same as the Imagenette dataset. We train a ResNet50 model from scratch with the same hyperparameter configuration using three different training methods.

1. DiSTinguish-Complete (**DiSTinguish-C**): 20-class supervised learning, the additional classes are the structure-disrupted version of the original class, and the structure-disrupted images are generated using Texture Synthesis with 100-step optimization.

2. DiSTinguish-Approximate (**DiSTinguish-A**): Similar to DiSTinguish-C, while the structure-disrupted images are generated using Texture Synthesis with only 10-step optimization.

3. Baseline: Simple 10-class supervised learning without any special augmentation.

We evaluate the above three methods on three different evaluation datasets: **DiST**, style-transferred version of the evaluation dataset of ImageNet10 (**SIN-10**) and original evaluation dataset of ImageNet10 (**IN-10**). Experiment results on ImageNet10 are shown in Table.3, all the models are ResNet50 trained within 100 epochs. Even though it doesn't reach the same performance as DiSTinguish-C, DiSTinguish-A still surpasses the baseline in both SIN-10 and DiST evaluations. This indicates that DiSTinguish-A serves as an effective approximation, particularly when it is impractical to generate DiSTinguish-C data on large-scale datasets.

## A.3  Model training detail and configuration

**Re-examine the models on DiST**  All the model we used during the evaluation on DiST and cue-conflict dataset are directly from the public pretrained models. ResNet50-SIN is the model trained on only stylized images in Geirhos et al. (2018). For the ResNet50 model we use the IMAGENET1K_V1 weights from pytorch. Others are the default pretrained weight.

Table 3: DiSTinguish on ImageNet10 (Top-1 Accuracy)

|               | IN-10 ($\uparrow$) | SIN-10 ($\uparrow$) | DiST ($\uparrow$) |
|---------------|--------------------|---------------------|-------------------|
| DiSTinguish-C | 93.2               | 71.6                | 95.5              |
| DiSTinguish-A | 93.2               | 70.0                | 88.4              |
| Baseline      | 90.2               | 54.4                | 54.9              |

**Experiment on ImageNet10**   In the small-scale experiment of ImageNet10, the class we select are exactly the same as the Imagenette dataset. The class label of the selected class are *n01440764*, *n02102040*, *n02979186*, *n03000684*, *n03028079 n03394916*, *n03417042*, *n03425413*, *n03445777*, *n03888257*. The model is trained by using SGD as the optimizer, with learning rate of 0.05, batch size of 256 on a single Tesla V100 GPU for 100 epochs. To eliminate the impact of data augmentation, no special augmentation is appiled during the experiment.

**Experiment on ImageNet1K**   The model we used in the experiment of ImageNet1K, except for the baseline model, which directly using the IMAGENET1K_V1 weights from pytorch, are trained from scratch using ffcv (Leclerc et al. (2023)). All the models are trained for 90 epochs with the start learning rate of 0.1 on a signle V100 GPU. Other configuration remains the same as the default configuration in ffcv for training ResNet50.

### A.4 PSYCHOPHYSICAL EXPERIMENT DETAIL

Psychophysical experiments are conducted using a front-end web application developed in JavaScript. Subjects are instructed to "Find the image that is different from the other two" and can select their answers using keys '1', '2', or '3'. After making a selection, subjects press the spacebar to proceed to the next question.

The trial procedure is illustrated in Fig.8. A set of images appears on the screen after a 300 ms delay and remains visible for 800 ms. In a standard trial, two structure-disrupted images and one original image are presented; the correct answer is the original image. Following the 800 ms display period, the images vanish, and subjects have an additional 1200 ms to make their selection, totaling 2 s for decision-making. If no selection is made within this time, the trial is marked as a timeout, and the response is considered invalid. Subjects are given the opportunity to take a break after every 100 images. To prevent the supervision signal, no feedback on answer correctness is provided during the test.

To mitigate the risk of the "oddity pop-out" test devolving into a mere "detection task"—where subjects might focus solely on identifying the original image rather than the one that differs—we incorporate extra catch trials into the experiments, as illustrated in Fig.9.

One catch trial is presented after every 10 standard trials. In each catch trial, two "original images" are displayed: one is a mirrored version of the other, accompanied by a structure-disrupted image. It is important to note that there is no overlap between the images used in catch trials and those used in standard trials. In these catch trials, the correct answer is actually the structure-disrupted image. The rationale for incorporating such catch trials is to compel subjects to focus on identifying the "different" image rather than the "original" one, thereby aligning the task more closely with how deep learning models behave during DiST evaluation. Results from the catch trials are not included in the final performance metric.

### A.5 MORE VISUALIZATION ANALYSIS FOR FOUR TRAINING APPROACHES

In this section, we present the original sensitivity maps without binary mask for a ResNet50 model trained using the four different approaches examined in our ImageNet1K experiment. Sensitivity maps serve to illustrate the model's responsiveness to pixel-level changes; a lighter pixel suggests a more significant influence on the model's decision-making process.

As shown in Fig.12, in line with the findings detailed in the main text, the sensitivity map of a model trained with both DiSTinguish and stylized augmentation qualitatively demonstrates the synergistic

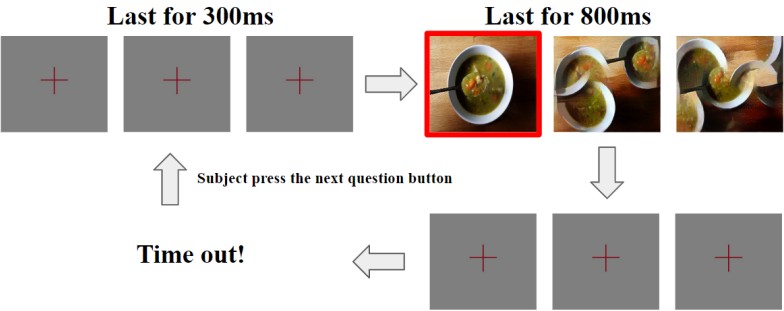

Figure 8: Standard trial in the psychophysical experiment. Image in the red box is the correct answer.

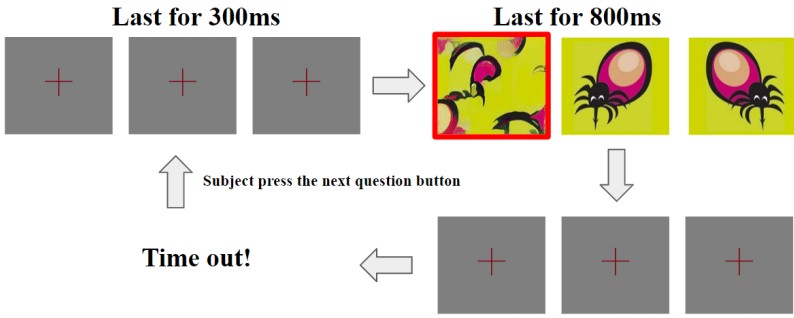

Figure 9: Catch trial in the psychophysical experiment. Image in the red box is the correct answer

effect of these methods. Specifically, the model learns to focus on specialized local features that are robust to style transfer, while also becoming attuned to the global structure of the object.

We further examine the feature vectors from the last layer of a ResNet50 model trained under different conditions by visualizing them using t-SNE. As illustrated in Fig.10, both the baseline model and the one trained with Stylized Augmentation fail to effectively distinguish between original and disrupted structure. Their corresponding classes in the feature space show significant overlap. In contrast, the model trained with DiSTinguish clearly separates feature clusters corresponding to original structure from those of their disrupted versions. And combining two methods together won't influence such separation.

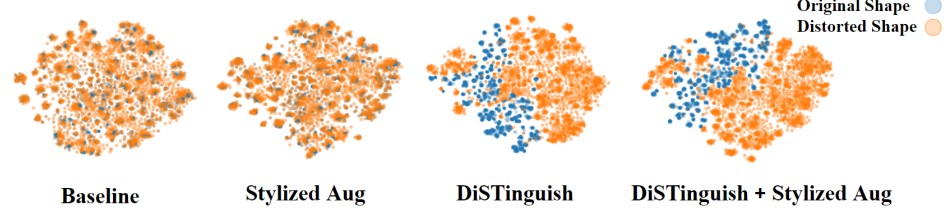

Figure 10: t-SNE visualizations of the feature vector of ResNet50 trained under different approaches.

## A.6    MORE EXAMPLE IN DIST

In this section we will show more example in DiST. Fig.11 shows 12 sets of images in DiST, each set consists of one original image (leftmost one) and two generated structure-disrupted image. Ideally,

we would want the optimization process to generate the image that the local components of the object are disrupted, to test if the model is sensitive to such change of the global shape. But depending on the characteristics of the original images, a small proportion of the generated results can be particularly hard for both humans and models. For example, results in Fig.11 (e) and Fig.11 (k) are the challenging cases, where the objects and background are difficult to distinguish. Those challenging cases might be less meaningful to evaluate the sensitivity of the global shape of the models. Even the dataset does include some of those cases, most of the content still follow our intent. And human is still able to distinguish most of the images in the dataset (over 85%), which outperform all the model without training with DiSTinguish, showing that such gap about the perception of global shape does exist.

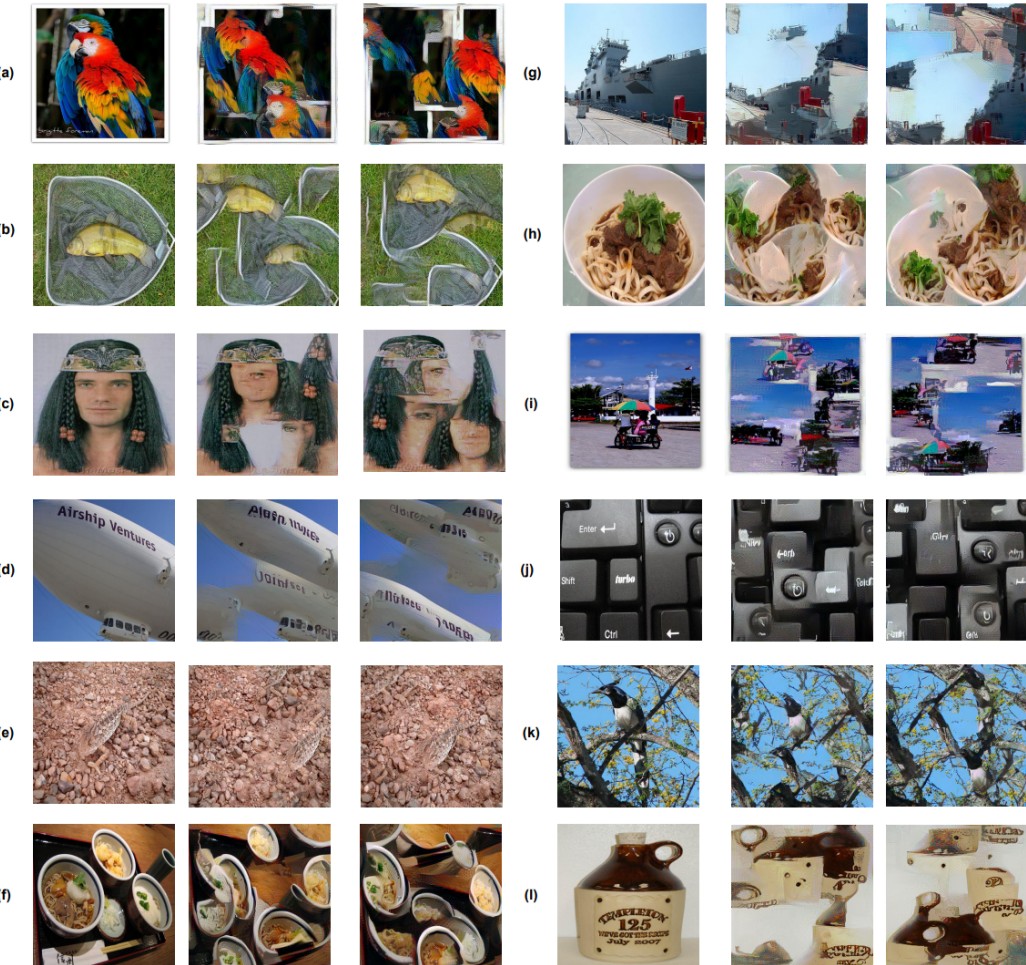

Figure 11: More example of disrupted structure images and its original images.The first image in each image set is the original one.

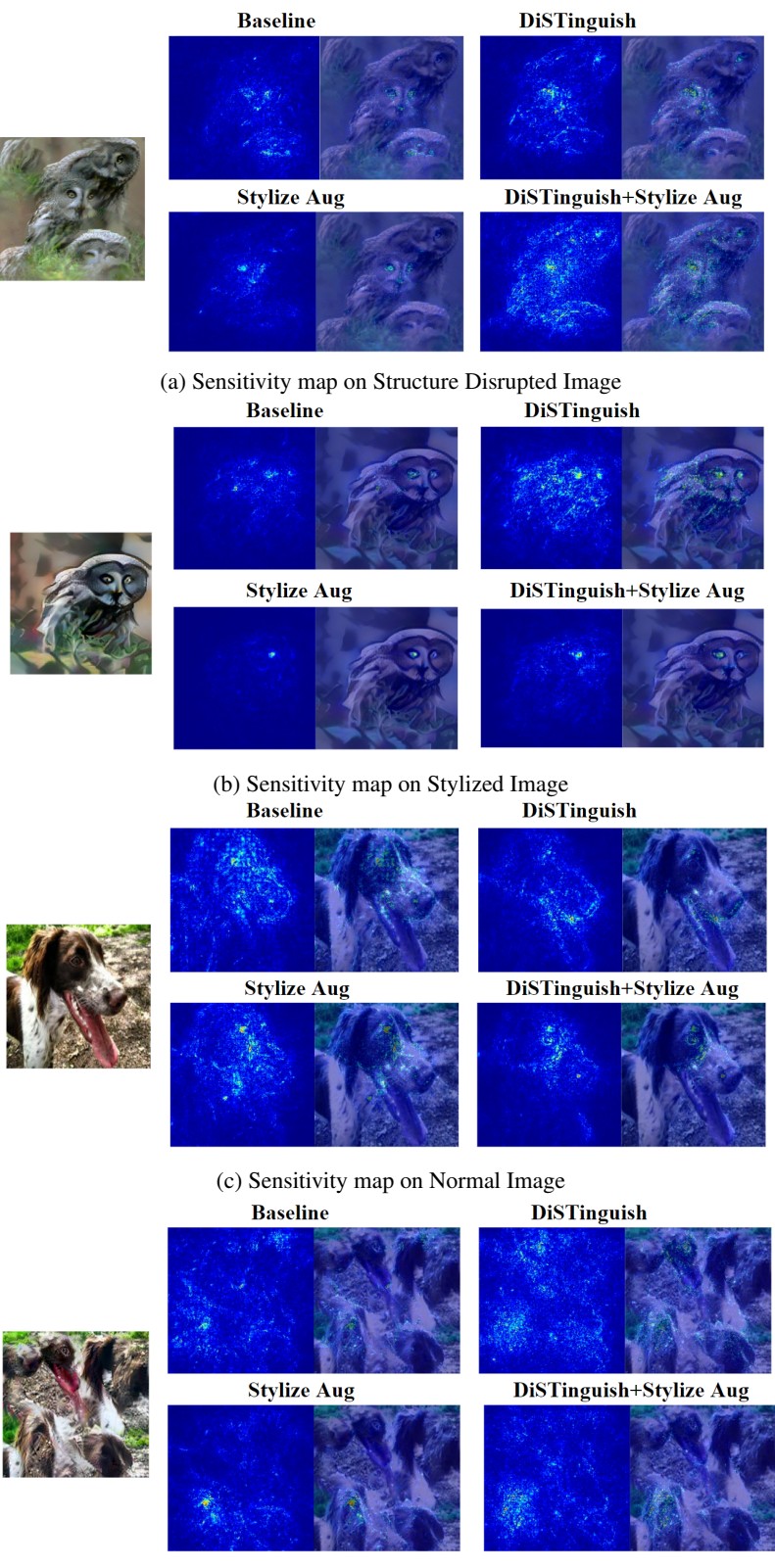

(a) Sensitivity map on Structure Disrupted Image

(b) Sensitivity map on Stylized Image

(c) Sensitivity map on Normal Image

(d) Sensitivity map on Structure Disrupted Image

Figure 12: Sensitivity map of ResNet50 trained under different methods, the lighter the point is, the stronger that pixel would influence the decision of the model.

