# OpenReview forum: "Does resistance to style-transfer equal Global Shape Bias? Measuring network sensitivity to global shape configuration"
_ICLR.cc/2024/Workshop/Re-Align — ICLR 2024 Workshop Re-Align Poster_

### Official Review · Reviewer_hYTo · 2024-02-24
**Global structure sensitivity**

**Rating:** 3
**Fit:** 3
**Confidence:** 2

**Workshop Review:**

This paper studies machine learning models’ understanding of global shape structure. The first contribution of the paper is a test bench (DiST) as a direct measurement of global structure sensitivity. The authors then conduct extensive experiments to evaluate various models on this benchmark, to examine vision transformer and masked autoencode’s ability to learn global structure, and to show that global structure sensitivity is orthogonal to resistance to style transfer.

Several findings in this paper are original and significant, in particular the independence of a model’s reliance on local textures and on global structures, and the failure of ViTs on the proposed benchmark. The experiments are comprehensive. The psychophysical experiment is convincing in quantifying the gap between deep learning models and the human visual system. The writing is very clear.

I did not find significant weaknesses in this paper. However, below are some questions that might help clarifying a few details:
- Does the DiST metric satisfy the triangle inequality, which is commonly required for distance measures?
- Why is learning global structures important? Is it correct to say that models that understand global structures well are closer to human reasoning?
- There is a repeat sentence in the conclusion. (Not relying on …)

**Reason For Not Giving Higher Score:**

N/A

**Reason For Not Giving Lower Score:**

Novel perspective and findings, as well as impressive experiment results.

**Reviewer Domain:**

machine learning

---

### Official Review · Reviewer_X1bS · 2024-02-27
**Relevant dataset and empirical study on human and model robsutness towards global shape bias**

**Rating:** 2
**Fit:** 3
**Confidence:** 3

**Workshop Review:**

The paper focuses on a relevant and interesting problem of deep learning architectures for computer vision, which is global shape bias. The paper contrasts global shape bias with resistance to style-transfer capabilities and thus is largely inspired by previous works [1,2]. I see its main contribution in the global shape dataset and its generation approach. Additionally, the human evaluation and model comparison of cue-conflict and DiST-Accuracy are interesting and relevant. I think the organization and clarity of the paper are good. I think the paper would benefit from a more in-depth discussion section to contextualize their findings and to gain more conclusive insights. There are a number of typos and grammatical errors that need to be fixed, e.g. on page 1: model understand -> models understand, that belong -> that belongs.

 [1] Robert Geirhos, et al.. Imagenet-trained cnns are biased towards texture; increasing shape bias improves accuracy and robustness. arXiv preprint arXiv:1811.12231, 2018.

 [2] Robert Geirhos, e al.,. Partial success in closing the gap between human and machine vision. Advances in Neural Information Processing Systems, 34:23885–23899, 2021.

**Reason For Not Giving Higher Score:**

No direct evidence of how much the model relies on global shape, e.g. measure of how much model attribution falls on global shape vs local shape or texture. The authors rely on showing a few samples of a sensitivity analysis (Figure 7), but no quantitative experiments are shown for this, which would have clearly improved the insight value of the paper.

Their secondary contribution DiSTinguish (training models using their proposed DIST dataset) lacks novelty since it simply adds more training data from a different data distribution. But I do agree it is an interesting comparison study, it does not provide a substantial methodological contribution.

Some of the most interesting empirical experiments (Section 4.3, Table 2) in which the benefits of augmented standard training data with stylized and/or DIST are evaluated, are only performed on ResNet without providing any justification. This makes it difficult to draw broader conclusions also for other model classes.

They provide evidence that Cue-Conflict scores do not give clear answers to which features are actually used, while introducing DiST to highlight the performance on global shape bias. Eventually, it seems that any of the metrics is not robust against possible shortcut solutions, since they all rely on evaluations of the predictions (instead of an evaluation of the features used by the model, e.g. using XAI methods to explain predictions [1] or similarity scores [2]). I think this point should be discussed more thorougly.

[1] W. Samek, et a., "Explaining Deep Neural Networks and Beyond: A Review of Methods and Applications," in Proceedings of the IEEE, vol. 109, no. 3, pp. 247-278, March 2021, doi: 10.1109/JPROC.2021.3060483.

[2] O. Eberle, et al., "Building and Interpreting Deep Similarity Models," in IEEE Transactions on Pattern Analysis and Machine Intelligence, vol. 44, no. 3, pp. 1149-1161, March 2022, doi: 10.1109/TPAMI.2020.3020738.

**Reason For Not Giving Lower Score:**

I think the Inclusion of a human evaluation study to measure how model and human ratings of global shape compare is relevant for the workshop.

The author’s provide an interesting dataset DiST and its generation approach to the community, which may be useful for future studies to investigate model robustness.

The´ evaluation of DIST performance was done over a wide selection of models and offers a good starting point for future work.

The gram-based generation approach to break global shape is a relevant contribution.

**Reviewer Domain:**

machine learning

---

### Official Review · Reviewer_WfgD · 2024-02-28
**Interesting new benchmark for shape bias but presentation needs improvement**

**Rating:** 2
**Fit:** 2
**Confidence:** 2

**Workshop Review:**

- Clarity: The paper’s aims are clear, but the results and details of the methods are not always fully clear and do not always meet standard of reporting. The paper contains a lot of grammatical and spelling errors.
- Correctness: As far as I can tell the analyses and results are correct, though not many details of methods are easy to find (e.g. how many images are actually used in DIST evaluation and training).
- Novelty: The paper presents a new benchmark to evaluate global shape sensitivity in humans networks using a pre-existing method from Gatys et al., 2017. It presents a suite of new analyses showing how this new benchmark demonstrates that decreasing texture bias in neural networks does not automatically increase global shape bias.
- Interest to the community: Given the general interest in texture bias as a well-known behavioral gap between DNNs and humans, the paper will be of some interest to this community. However, the paper focuses less on the human side (besides providing a novel accuracy benchmark to achieve, the human data is not further analyzed) and more on the DCNN modeling side, trying to understand why the models fail to represent global shape and what architectures and training objectives can be used to increase this representation.

**Reason For Not Giving Higher Score:**

- Incomplete reporting of psychophysical experiments (e.g. subject age and gender is not reported, number of trials/subject are not reported, experimental settings such a screen size and image resolution are missing). Variation in accuracy (SD or SEM) across subjects or images is lacking.
- It’s not clear why the new training paradigm (DiSTinguish) would be needed if ViT-MAE already achieves super-human performance on the new DIST benchmark.
- Although the paper contains data from two systems (human and deep networks), the analyses and focus seems more pure ML (focusing on beating the benchmark). There is not discussion about whether MAE is 'sensible' from a human cognition perspective for example.
- It would be good if the paper had discussed the results reported in Fig 5 elaborately, for example why the 'sparse' models appear to be doing the best in terms of approaching human performance on DIST.
- It would be interesting to include more detailed assessment of the human performance, e.g. which specific images or classes are more difficult for humans to distinguish.
- The presentation can be much improved in clarity, both structurally (for example relevant figures are in Appendix but cited in main text) and writing (many grammatical errors).

**Reason For Not Giving Lower Score:**

- The paper focuses on the texture and shape bias debate which is a much-studied example in the literature on (mis)alignment of humans vs. deep networks.
- Most of the graphs are insightful and effectively illustrate why the most prominent technique to alleviate texture bias in deep networks (stylized augmentation) does not increase global shape representation.

**Reviewer Domain:**

neuroscience

---

### Decision · Program_Chairs · 2024-03-02

Accept (Poster)